# The Influence of Reducing Clinical Practicum Anxiety on Nursing Professional Employment in Nursing Students with Low Emotional Stability

**DOI:** 10.3390/ijerph19148374

**Published:** 2022-07-08

**Authors:** Mei-Hsin Lai, Chyn-Yuan Tzeng, Hsiu-Ju Jen, Min-Huey Chung

**Affiliations:** 1School of Nursing, College of Nursing, Taipei Medical University, 250 Wu-Hsing Street, Taipei 11031, Taiwan; g8805001@sunrise.hk.edu.tw; 2Department of Nursing, HungKuang University, No. 1018, Sec. 6, Taiwan Boulevard, Shalu District, Taichung 433304, Taiwan; 3Taiwan Home Care & Service Association, Room A1415H, Medical Building, 250 Wu-Hsing Street, Taipei 11031, Taiwan; tcy9907@gmail.com; 4Department of Nursing, Shuang Ho Hospital, Taipei Medical University, New Taipei City 23561, Taiwan; 15033@s.tmu.edu.tw

**Keywords:** low emotional stability, clinical practicum anxiety, nursing professional employment, coping, adapting strategies, hospital nursing units

## Abstract

Nursing students experience anxiety during clinical practicum, which may interfere with their learning in clinical practice and nursing employment after graduation. This study explored: (1) the factors of the difference in anxiety levels between pre- and post-practicum in nursing students; (2) identified their anxiety events in a clinical environment; and (3) the correlation between emotional stability and 5-year nursing professional employment. The study was designed as a mixed method. A longitudinal secondary analysis method and a qualitative approach with open questionnaire were conducted. The emotional stability subscale of Lai’s Personality Inventory and the Beck Anxiety Inventory, as well as open questionnaires were administered. Research data were collected through the purposive sampling of 237 nursing students (mean age was 20.96, SD = 1.29) of 4.2% male and 95.8% female in a central Taiwan hospital in 2013, and the participants were followed up in 2021 to show 70% in clinical service. Most of the nursing students exhibited significantly decreased anxiety levels in the post-practicum period. Compared to nursing students with high emotional stability, those with low emotional stability exhibited higher differences in their anxiety levels between the pre- and post-practicum periods. Low emotional stability is critical in a pre-practicum BAI score. However, the high pre-practicum BAI score would decrease to normal range after enrolling to practicum setting. So, as to their 5-year nursing professional employment after graduation. Teachers foster a positive learning atmosphere that emphasizes the importance of “we are family” to students. Teachers and advisors need to make efforts in leading the low emotional stability nursing students to learn effective coping and adapting strategies in clinical practicum.

## 1. Introduction

The nursing practicum is critical to nursing students, because it provides them with opportunities to apply their knowledge and skills to clinical nursing [1,2]. Although clinical practice is an essential part of nursing education, many students experience negative emotions, such as anxiety, when performing in real-life situations. The high levels of anxiety experienced by nursing students are a concern internationally because anxiety can interfere with learning by affecting the performance of nursing studies and their ability to focus during their clinical practicum [3,4,5].

Anxiety events refer to the sources that cause the nursing students to experience anxiety. Based on a review of the literature, the anxiety events experienced by the nursing students in clinical environments can be classified into three categories: difficulty in adapting to the medical team; difficulty in performing the nursing processes; and difficulty in performing nursing skills [5,6,7,8,9,10,11,12]. The anxiety event of difficulty in adapting to the medical team is a result of the stressors that are contributed to by professionals, such as the clinical teachers and nursing staff, as reported by the researchers [6]. The clinical teachers usually focus on the negative aspects of the performance of the nursing students, which is a crucial cause of stressful relationships between the clinical teachers and nursing students [7]. In addition, the nursing students may encounter the stressor of an unsupportive environment, which is mainly contributed to by the nursing staff [8]. The event of difficulty in performing the nursing processes can be attributed to the feelings of uncertainty that the nursing students experience when providing nursing care to patients, especially when encountering a patient with various diseases or patient death [9]. This uncertainty is due to their lack of professional proficiency and the highly challenging variety of diseases [10]. Furthermore, the event of difficulty in performing nursing skills can be attributed to the fears that the nursing students have of making mistakes [11] that may cause them to harm patients [10], or to respond to patients inappropriately [12], especially during their initial clinical experience in the unit, where they may be concerned about poor time management and being late [10].

Emotional stability is the process by which one’s personality continuously strives for a greater sense of emotional health, both intra-physically and intra-personally [13]. Emotionally stable people can tolerate many situations: they can withstand delays in satisfying their needs; tolerate a reasonable amount of frustration; trust their long-term planning capacity; and accept delays in their expectations and needs, or revise them as necessary [13]. Emotional instability is associated with anxiety [14]; hence, it is an essential variable in anxiety research.

There were associations between emotional stability and anxiety. As Akram et al. [15] mentioned, insomnia is associated with conscientious and emotional stability. Anxiety was a mediator of emotional stability. Ahn et al. [16] studied the relationship and said that low emotional stability would affect sleep and was mediated by anxiety. Ahmed et al. [17] outlined that anxiety was not significantly related to gender and age, but was negatively and significantly correlated to emotional stability.

Saad et al. [18] concluded in their study that emotional stability was positively, highly significantly related to the job crafting of nurses. İspir et al. [19] in their study showed that there was a correlation between personality traits and career adaptation. Dilmini and Thalgaspitiya’s [20] study on the relationship between personality and job engagement found that there was a positive moderate relationship between the personality and job engagement of employees. They found that conscientiousness was the most determinant personality trait for job engagement and suggested that low emotional stability was good for employees who worked in stressful situations.

Serebryakova et al. [21] mentioned that emotional stability was an outcome of social–psychological adaptation, through the process of effective accustoming to the social environment, acceptance of the standards of environment, to reach an emotionally stable condition. So, adaptation strategies are important for students and nurses.

Regarding practicum anxiety, many of the studies have investigated the anxiety experienced by nursing students during the pre- and post-practicum periods [22]. Some studies have explored the differences in stress and self-esteem [23] and self-perception [24] between pre- and post-practicum nursing students. In addition, a study assessed how a structured learning program affected the students, both before and after the program [25]. A positive difference in the scores indicated that the nursing students experienced anxiety in the initial period, but less anxiety as the practicum progressed [4].

Therefore, this study aimed to find if the emotional stability of the nursing students can affect the clinical practicum anxiety and the nursing employment 5 years after graduation, including the condition and their coping strategies. The purpose of this study included: (1) the factors of the difference in anxiety levels between pre- and post-practicum in nursing students; (2) identified their anxiety events in a clinical environment; and (3) the correlation between emotional stability and 5-year nursing professional employment.

## 2. Materials and Methods

### 2.1. Study Design

This study was designed as a mixed method study. A longitudinal secondary analysis method was used to describe the condition of emotional stability, the clinical practicum anxiety, and the 5-year nursing professional employment after graduation. A qualitative approach with open questionnaires was used to explore the adaptive strategies of the nursing students.

### 2.2. Study Sample

We applied a secondary analysis method in this study. The secondary analysis dataset population was comprised of 237 nursing students. The first analysis study population was comprised of 245 nursing students, practicing in a central Taiwan hospital. The first analysis study was posted in 2015. Purposive sampling was adopted in this study, according to the following criteria: (1) the participants had reached the last week of their practicum period; (2) the participants were able to write; and (3) the participants were willing to share their clinical nursing practicum experiences, particularly those related to anxiety events. The practicum program lasted for 3–4 weeks. During the practicum period, about six–seven students were assigned to a nurse teacher. The frequency and duration of the different practicums are as follows: (1) medical or surgical internships last for two months; (2) gynecology internships last for 3–4 weeks; (3) pediatric internships last for 4 weeks; (4) public health internships last for 3 weeks; and (5) psychiatry internships last for 3 weeks. All of the students attended their practicum placements 5 days a week. The public health students completed their internships in a hospital.

Because the original study was formed to survey the condition of all of the nursing students in a central Taiwan hospital in 2013, the data from the first study were only suitable for evaluating the hospital in the first study. As such, convenience sampling was used.

The required sample size was estimated using G*power 3.1 software [26]. As per a priori power analysis, an R^2^ value of 0.135 was used to determine the effect size (the obtained value was 0.16) with an alpha value of 0.05, the number of factors set at 14, and the power set at 0.95. The estimated sample size required for linear multiple regression was 187 participants. After adding 20% to account for the invalid samples, the total number of participants required for this study was 225.

### 2.3. Instrument

#### 2.3.1. Emotional Stability

The emotional stability subscale of the Mandarin version of Lai’s Personality Inventory (LPI) was developed and validated by Lai and Lai [27]. The Mandarin version of the LPI comprises 15 subscales measuring four personality traits (introversion–extraversion, emotional stability, psychological health, and good social adaptation–bad social adaptation). Cyclic tendencies, feelings of inferiority, and nervousness indicate one’s level of emotional stability. For each subscale, a score of one or two represents high emotional stability; a score of three represents medium emotional stability; and a score of four or five represents low emotional stability. The LPI possesses good reliability (test–retest reliability ranges from 0.71 to 0.93 across all of the subscales) and validity (concurrent validity ranges from 0.62 to 0.81), and it is widely used in Taiwan.

#### 2.3.2. Beck Anxiety Inventory

The Beck Anxiety Inventory (BAI), which was developed by Beck [28] and translated and validated by Che et al. [29], was used in this study to check for anxiety symptoms in the sample. This inventory is a self-report instrument with 21 multiple-choice items related to four factors: neurophysiological; subjective; panic; and autonomic symptoms. The BAI scores range from 0 to 63; a score of <7 represents a “minimal” level of anxiety; a score of 8–15 represents “mild”; a score of 16–25 represents “moderate” anxiety; and a score of 26–63 represents “severe” anxiety. The Chinese version of the BAI possesses high internal consistency (Cronbach’s α = 0.95; Guttman split-half coefficient = 0.91), and a factor analysis revealed a two-factor structure. The total variance in the Chinese version of the BAI was 58.04% and was similar to Beck’s original construct.

#### 2.3.3. Anxiety Events during Clinical Practice

According to the goals of the present study, the participants were asked open questions, such as “please describe your current anxiety events during nursing practicum”. Data on the anxiety events of “difficulty in adapting to the medical team”, “difficulty in performing the nursing processes”, and “difficulty in performing nursing skills” were analyzed, using the content analysis method. Then, the number of times that each theme occurred in the questionnaire responses submitted by the nursing students was quantified. To ensure accuracy, two separate raters coded the themes and then quantified them [30,31].

#### 2.3.4. The 5-Year Nursing Professional Employment

In this study, the indicator of clinical performance and retention used was a 5-year nursing professional employment. The 5-year nursing professional employment means that the professional nursing employment unit was a setting that needed nursing performance, such as the hospital nursing units for medical and surgical, obstetrics, pediatrics, psychology, the community unit of a medical center, or a regional hospital. We followed up on the participants’ jobs by reviewing the dataset in school in 2021.

#### 2.3.5. Coping Process during Clinical Practice

The participants were asked to answer the questions “please describe your resources for current anxiety events during nursing practicum”.

### 2.4. Procedure

The pre-practicum questionnaire was applied on the first day 8 a.m. of practicum before entering the ward. The post-practicum questionnaire was applied on the last 3 days of the last week of practicum. All of the participants received an oral explanation of the study purpose and signed an informed consent form. Thereafter, the research instrument was administered. The participants were also assured that absolute confidentiality would be maintained. The convenience and privacy of the setting were considered for the safety of the participants. The investigator primarily selected centers where the students conducted their practicum placements as the study location, and investigated the students without the involvement of any of the other students or members of the nursing staff. To express our gratitude to the participants who supported this study and to make them more willing to participate, each participant was rewarded a coupon (TWD 50-dollar) after completing the questionnaire.

### 2.5. Ethical Considerations

This study was approved by the Institutional Review Board of a General Hospital (approval number: people 10145).

### 2.6. Data Analysis

The collected quantitative data were coded and imported into the SPSS version 26.0 for Windows. The data were expressed as frequencies and percentages for categorical variables and means and standard deviations (SDs) for continuous variables. Paired *t*-tests, simple linear regression, and hierarchical multiple linear regression were performed to explore the relationship between emotional stability and the difference between pre- and post-practicum anxiety levels in nursing students. The qualitative data were used for content analysis to explore the adaptive strategies during the clinical practicum of nursing students.

## 3. Results

Table 1 summarizes the characteristics of the nursing students. The mean age was 20.96 (SD = 1.29), and most of the participants were aged between 20 and 26 years old (72.2%). Most of the participants were women (95.8%), had received a 2-year Bachelor of Science for nursing students (46.8%), and practiced in a medical or surgical setting (50.6%). The percentage of participants with previous practicum experience represented slightly more than half of the participants. Regarding the distribution of emotional stability levels, 20.3%, 48.9%, and 30.8% of the participants exhibited high, medium, and low levels of emotional stability, respectively. Almost half of the participants experienced difficulty in adapting to the medical team (48.9%). The participants’ pre-practicum and post-practicum BAI scores ranged from 0 to 39 and from 0 to 29, with mean scores of 5.66 (SD = 6.23) and 3.98 (SD = 5.32), respectively. In addition, compared to the pre-practicum scores, most of the mean of BAI scores decreased significantly during the post-practicum period, with the exception of the participants who were male, who were working in an obstetric or community practice setting, who had a high or medium level of emotional stability, or who were experiencing the anxiety event of difficulty in performing the nursing processes. Most of the nursing students had a BAI score of <8 in the pre-practicum period (75.1%), and the percentage of nursing students with a BAI score of <8 increased during the post-practicum period (81%). Nearly a quarter of the nursing students (24.9%, *n* = 59) had a BAI score of >8 during the pre-practicum period; however, the percentage of nursing students with a BAI score of >8 decreased during the post-practicum period (19.0%, *n* = 45).

Simple and multiple linear regression analyses were conducted to assess the factors affecting the differences in the BAI scores between the pre- and post-practicum periods (Table 2). A simple linear regression revealed that emotional stability was a significant factor affecting the difference in the BAI scores between the pre- and post-practicum periods, particularly for low emotional stability (β = 0.27, *p* = 0.001) compared to high emotional stability. To determine the effect of emotional stability, the demographic variables in Table 2 were controlled and revealed that the nursing students with low emotional stability (β = 0.29, *p* = 0.001) were more likely to attain a significant difference in their BAI scores between the pre- and post-practicum periods, compared to those with high emotional stability. The results indicated that the factors in Model 3 accounted for 13.5% of the total variance (R^2^ = 0.135, *p* = 0.003) in the differences in the BAI scores between the pre- and post-practicum periods.

In Table 2, the results showed that low emotional stability was the influencing factor on the BAI scores. This finding, combined with Table 1, showed that the pre-practicum mean BAI score of the low emotional stability students decreased from 8.97 to 4.44 after enrolment to the clinical practicum. The post-practicum mean BAI score of the low emotional stability participants, 4.44, was similar to the medium emotional stability students (pre- and post- practicum mean BAI scores were 4.49 and 4.32).

Table 3 shows the number and distribution of the in-service participants and the 5-year nursing professional employment, according to the emotional stability level, pre-BAI score, and post-BAI score. The 5-year nursing professional employment conditions included service in the hospital nursing units for medical and surgical, obstetrics, pediatrics, psychology, the community unit of a medical center, or a regional hospital in 2021. The 5-year college students were excluded because they were about to continue their school learning and become a 2-year BSN student after graduation.

In Table 3, we found that the low emotional stability participants had a 37.10% of 5-year nursing professional employment, instead of the medium emotional stability participants 42.42%, and the high emotional stability participants 36.57%. We found that low emotional stability did not significantly affect the 5- year nursing professional employment. The reason might be the findings in Table 1, where the pre-practicum anxiety score was decreased from 8.97 to 4.44 after entering the clinical setting.

In Table 4, we ruled out the estimating criteria of the 5-year nursing professional employment among the low emotional stability participants. There were 23 low emotional stability participants who did not fail in the 5-year nursing professional employment. So, we analyzed their statement about the adaptive strategies that these participants learned in clinical practicum. The main theme was that “we are family”. The teacher used a variety of techniques to foster a helpful learning environment throughout the nursing practicum. The students were able to practice in a nurturing learning setting where they felt like a family and were taught coping mechanisms for anxiety. The following offers some expressions on the subject.

Firstly, we saw that the students who were originally predicted to be a ‘failure; could not adapt anxiety after enrolling clinical setting’, were 42.86% not a failure. The important points expressed by the students included: “guided and lead me”; “encouraged me”; “good teamwork model example”.

For example, one participant said: “My teacher constantly gave me encouragement. We behaved just like a family during the practicum”. (No. 212). The environment for the practicum was positive.

Second, 38.89% of the participants who were predicted to be “not failure” after entering in a clinical environment were, in fact, not a failure. Seven people participated. The key points raised by the students included: “guide me/direct me”, “help me”, “display and explain”, “use multimedia”, “discuss the anxiety problem”, “offer an opinion of stress relief method”, “encourage me”, “be patient”, and “give chance”.

For example, one participant said: “Teacher taught me how to work more quickly and helped me care for patients”. (No. 4). “Demonstrated and explained how to better in my jobs, utilized multimedia to help”. another participant said (No. 10). The third participant stated: “Talking to me about my anxious problem. Share my thoughts on this form of stress alleviation”. (No. 13). “Encouraging me”, the fourth participant stated, “being patient with me and assisting me in solving the issues”. (No. 49). “Encouraging me and leading me, being patient with me and giving me a chance”, the fifth participant stated (No. 127). The sixth individual (No. 174) remained silent the entire time. According to the seventh participant, “Assisted me in solving the situation”. (No. 236).

Thirdly, there was a 38.24 percent success rate among the students who were predicted to be “Trace; may adapt anxiety after enrolling in clinical setting”. There were thirteen people present. In addition to “immediately paying attention to my problems”, “leading me to thinking”, “leading me in my direction”, “did not be too serious”, “in patience”, “very nice”, “our relationship was very good”, “taught me to relax, open my mind, do not be stubborn, just try my best to do”, “nice communication with students”, “gave support”, “understood the cause of anxiety and led me how to exclude it”, “chatted with me about my anxious”, and “like a family”, “served as a conduit between me and the practice environment”, “slowed down the pace”, “encouraged me”, “improved my confidence”, and “instructor led me in the direction of problem solving”.

One participant, for instance, stated: “Whenever I had an issue and communicated with the teacher, the teacher was always attentive to my problems. Additionally, the lead nurse in the practicum was”. (No. 1). “Teacher would inspire me to thinking about which would be best and what would be my direction”, a different participant remarked (No. 7). “Teacher led me in my direction”, remarked the third participant, “was not very serious”. (No. 22). The fourth participant remarked, “The teacher was really kind and patient with us, and our relationship was excellent”. (No. 36). “Teacher helped me to relax and expand my thoughts”, the fifth participant stated, “Be not obstinate. Just give it my all, and let fate take its course”. (No. 126). “Nicely communicated with students, did not get furious or give pupils attitude”, the sixth participant said, “shown support as opposed to apathy”. (No. 135). “Teacher realized the cause of my fear and showed me how to exclude it”, the seventh participant commented, “My teacher and I talked about my anxiety, but she didn’t dwell on it too much to prevent making me feel worse”. (No. 139). The eighth participant (No. 176) remained silent the entire time. “The teacher offered me several responses in my reflection”, the ninth participant stated, “We felt at home and content with the practicing environment”. (No. 213). “Teacher served as a conduit between me and the practice situation”, the tenth participant stated, “My teacher showed me how to prepare my task”. (No. 215). “Teacher lowered the speed for me and encouraged me”, the eleventh participant said (No. 219). The twelfth participant stated: “Teacher gave me confidence and encouragement”. (No. 235). The thirteenth participant stated: “My teacher guided me toward problem-solving. My teacher is supportive rather than critical. Once the teacher had demonstrated for me, I had followed their instructions to operate”. (No. 240).

According to all of the statements made by the students above, we can conclude that the teacher taught the students how to adapt to their surroundings.

## 4. Discussion

In this study, we found that low emotional stability was the influencing factor on the BAI score and was consistent with Akram et al. [15], Ahn et al. [16], and Ahmed et al. [17].

The major findings of the present study are that the subjects with low emotional stability exhibited a significant decrease in their post-practicum BAI scores, and emotional stability was found to be a significant factor in the differences in the BAI scores between the pre- and post-practicum periods. Low emotional stability is identified as neuroticism or emotional instability [32]. According to Tackett and Lahey [33], people with low emotional stability are likely to interpret situations as being highly threatening; thus, nursing students with low emotional stability experience higher anxiety than those with medium and high emotional stability during their practicum placements. In our study, the positive motivational tendencies of the nursing students with low emotional stability tended to be inhibited at work, leading to poor performance because of the time spent worrying about their performance and abilities; to overcome this, they required assurance from other people. However, when the nursing students actually entered the clinical environment and interacted with the nursing staff, they felt less threatened in the post-practicum period. Conversely, the nursing students with high and medium emotional stability exhibited relatively low anxiety in the pre-practicum period (BAI score < 8), as well as in the post-practicum period.

Finally, through a 9-year follow up period, a higher percentage of the students experienced long-term clinical service. According to Friedman [32], low emotional stability may lead to poor performance because of the time spent worrying about performance and ability, which causes a loss of focus on the work that needs to be completed in the moment. In the present study, the reason for this may be that the nursing students with low emotional stability were still learners in practicing hospitals or clinical settings; therefore, they required more time to adapt to the steps involved in performing nursing skills. These were consistent with the results of Wang et al. [13] and Milic et al. [14].

Moreover, the nursing students who experienced difficulties when performing the nursing processes did not exhibit a significant decrease in their post-practicum BAI scores, which was consistent with the results of Wang [9], Wang and Lee [10], Kim et al. [11], and Lin et al. [12]. This may be due to the complex steps involved in professional nursing assessments, analyses, diagnoses, interventions, and outcome evaluations, all of which were some of the challenges faced by the nursing students who experienced difficulties when performing the nursing processes.

The positive value of the differences in the BAI scores between the pre- and post-practicum periods revealed that the post-practicum BAI scores decreased. This finding agreed with the point of view from Wang [4] and Saad et al. [18], who mentioned that nursing students’ anxiety level is higher in the initial period than in the follow-up period. If the practicum program lasted more than 3 weeks, the nursing students would adapt and their pre-practicum anxiety score might decrease to normal within 2–3 weeks after enrolling in the clinical setting.

Hence, we concluded that: (1) the influence on nursing performance of low emotional stability could be overcome within 2–3 weeks after enrolling in the clinical practicum setting. Through clinical training, the nursing students would become familiar with and adapt to the clinical environment and in the same way to their nursing professional employment after graduation; (2) The adaptive strategy used by the low emotional stability students worked within 2–3 weeks, and this finding was consistent with the results of Ispir et al. [19], as mentioned, that adapting to the environment was an important influencing factor on the low emotional stability participants’ successful engagement in the clinical nursing job.

A positive learning atmosphere supports the students’ health and wellbeing. We discovered that the conducive learning environment in this study was “we are like family”. Additionally, the pupils would acquire the supplementary techniques and use them in similar circumstances in the future. The teachers are crucial in helping students with low emotional stability adjust by fostering a supportive learning environment.

The crucial teaching behaviors included: “leading and guiding me”; “encouraging me”; “good teamwork model example”; “assisted me”; “demonstrated and explained”; “applied multimedia”; “discussing the anxiety situation”; “giving an opinion of stress relief method”; “being patient”; “giving chanc”; “immediately pay attention to my problems”; “did not be too serious”; “very nice”; “our relationship was very good” and “taught me to relax”; “nice communication with students”; “gave support”; “understood the cause of anxiety and led me how to exclude it”; “chatted with me about my anxious” and “like a family”; “served as a conduit between me and the practice environment”; “slowed down the pace”; “improved my confidence”.

Since the above findings in this study, we followed up on the 5- year nursing professional employment of the participants after graduation till 2021. After a 9-year follow up period, a larger proportion of students were engaged in clinical work. There was a higher percentage of students with low emotional stability who experienced long-term clinical service.

Although low emotional stability might affect the anxiety score before enrolling in the clinical practicum setting, once they had enrolled in the clinical setting, the nursing students might make efforts to use effective adaptive strategies to cope with anxiety, which led to a long- term adaptation to stable nursing employment performance. As Dilmini and Thalgaspitiya [20] mentioned, conscientiousness determined the adaptation to engagement in the job and suggested that the low emotional stability participants would become good employees in stressful job environments.

This study had several limitations. First, we only recruited nursing students from one educational institution and investigated their experiences during only one stage of their clinical practicum. Second, the sample only included a few male participants and nursing students who were completing their practicums in community settings. Third, the purposive sampling was adopted to recruit participants from a single institute. Fourth, we did not explore the issue of “as educational experience increases, do stress levels increase or decrease?” The study outcomes might not be able to be generalized to male participants, students in a community practicum, or other institutes. Therefore, further research is suggested as follows to: (1) explore different stages of internship; (2) recruit a larger sample in multi-center hospitals; (3) include more male nursing students; (4) quantify the qualitative findings related to the adaption strategies in this study.

## 5. Conclusions

This study provides evidence that the low emotional stability of nursing students is critical to managing their pre- practicum anxiety levels. However, the high pre-practicum anxiety score would decrease to a normal level after enrolling in the practicum setting with effective adaptive strategies, and so to their 5-year nursing employment after graduation. Moreover, the low emotional stability participants might need time to adapt to a new environment to cope with anxiety. So, they might not like change in their work and could achieve a long-term nursing profession employment. So, teachers and advisors need to make efforts in leading the low emotional stability nursing students to learn effective coping and adaptive strategies in the clinical practicum. Teachers and schools may build the feeling that we are family. In addition, interesting and healing lessons may be designed into the school curriculum to create a supportive learning environment, one example of which would be a pet therapy course.

## Figures and Tables

**Table 1 ijerph-19-08374-t001:** Characteristics of the study participants and distribution of the pre-practicum and post-practicum BAI scores (*n* = 237).

Variable	*n* (%)	BAI Score		
		Pre-Practicum	Post-Practicum	
		(Mean ± SD)	(Mean ± SD)	*p*-Value
All participants	237 (100)	5.66 ± 6.23	3.98 ± 5.32	0.000 ***
Age (years)				
18–20	66 (27.8)	7.12 ± 6.08	5.30 ± 6.02	0.015 *
20–26	171 (72.2)	5.09 ± 6.21	3.47 ± 4.64	0.001 ***
Gender				
Male	10 (4.2)	4.20 (7.51)	2.30 (3.50)	0.468
Female	227 (95.8)	5.72 (6.18)	4.05 (5.38)	0.000 ***
Nursing Program				
5-year college	35 (14.8)	9.26 (6.69)	6.77 (7.15)	0.017 *
4-year BSN	91 (38.4)	4.70 (6.10)	3.65 (4.86)	0.044 *
2-year BSN	111 (46.8)	5.31 (5.83)	3.37 (4.76)	0.004 **
Practice Setting				
Medical or Surgical	120 (50.6)	5.78 (6.59)	3.86 (5.42)	0.003 **
Obstetric	39 (16.5)	5.21 (5.69)	5.26 (5.50)	0.948
Pediatric	33 (13.9)	4.73 (4.16)	2.36 (3.09)	0.001 ***
Psychiatry	32 (13.5)	5.13 (6.18)	3.28 (5.48)	0.011 *
Community	13 (5.5)	9.54 (8.05)	7.08 (6.50)	0.359
Previous Practicum Experience				
No	116 (48.9)	4.96 (5.27)	3.60 (4.85)	0.027 *
Yes	121 (51.1)	6.33 (6.98)	4.34 (5.73)	0.000 ***
Emotional Stability				
High	48 (20.3)	3.44 (3.87)	2.46 (3.64)	0.092
Medium	116 (48.9)	4.49 (4.65)	4.32 (5.93)	0.732
Low	73 (30.8)	8.97 (8.14)	4.44 (5.11)	0.000 ***
Difficulty in Adapting to the Medical Team				
No	121 (51.1)	5.48 (6.71)	3.29 (4.91)	0.000 ***
Yes	116 (48.9)	5.84 (5.71)	4.70 (5.65)	0.037 *
Difficulty in Performing Nursing Processes				
No	151 (63.7)	5.54 (6.15)	3.60 (4.67)	0.000 ***
Yes	86 (36.3)	5.87 (6.40)	4.65 (6.28)	0.109
Difficulty in Performing Nursing Skills				
No	142 (59.9)	5.60 (6.25)	3.92 (5.34)	0.001 ***
Yes	95 (40.1)	5.75 (6.22)	4.06 (5.31)	0.014 *
Proceeding of Anxiety		*n* (%)	*n* (%)	
BAI < 8		178 (75.1)	192 (81)	
BAI ≥ 8		59 (24.9)	45 (19)	

* *p* ≤ 0.05; ** *p* ≤ 0.01; *** *p* ≤ 0.001; BSN, Bachelor of Science in nursing.

**Table 2 ijerph-19-08374-t002:** Comparison of the predictors of the professional performance of the participants (*n* = 237).

Variable	Simple Linear Regression	Multiple Linear Regression		
		Model 1	Model 2	Model 3
		R^2^ = 0.033, *p* = 0.377	R^2^ = 0.126 ***, *p* = 0.001	R^2^ = 0.135 **, *p* = 0.003
	β (95% CI)	β (95% CI)	β (95% CI)	β (95% CI)
Age	0.023 (−0.49–0.70)	0.09 (−0.40–1.25)	0.05 (−0.55–1.03)	0.05 (−0.59–1.02)
Gender				
Male	Ref.	Ref.	Ref.	Ref.
Female	−0.01 (−4.10–3.64)	−0.00 (−4.24–4.05)	−0.00 (−4.02–3.95)	0.02 (−3.59–4.48)
Nursing Program				
5-year college	Ref.	Ref.	Ref.	Ref.
4-year BSN	−0.12 (−3.81–0.95)	−0.18 (−5.41–0.89)	−0.13 (−4.57–1.47)	−0.09 (−4.18–2.04)
2-year BSN	−0.05 (−2.86–1.77)	−0.12 (−4.75–1.98)	−0.07 (−4.06–2.38)	−0.04 (−3.70–2.81)
Practice Setting				
Medical and Surgical	Ref.	Ref.	Ref.	Ref.
Obstetric	−0.12 (−4.18–0.22)	−0.13 (−4.53–0.27)	−0.11 (−4.02–0.58)	−0.11 (−4.11–.56)
Pediatric	0.03 (−1.91–2.79)	0.02 (−2.25–2.80)	0.02 (−2.12–2.70)	0.01 (−2.30–2.60)
Psychology	−0.01 (−2.46–2.30)	−0.04 (−3.42–1.91)	−0.004 (−2.62–2.50)	0.01 (−2.50–2.87)
Community	0.02 (−2.95–4.02)	−0.03 (−4.61–3.04)	0.02 (−4.10–3.21)	0.00 (−3.72–3.90)
Previous Practicum Experience				
No	Ref.	Ref.	Ref.	Ref.
Yes	0.05 (−0.91–2.19)	0.08 (−0.89–2.93)	0.09 (−0.73–2.92)	0.10 (−0.66–3.03)
Emotional Stability				
High	Ref.		Ref.	Ref.
Medium	−0.07 (−2.76–1.14)		−0.05 (−2.61–1.37)	−0.03 (−2.40–1.66)
Low	0.27 (1.44–5.67) ***		0.27 (1.43–5.73) ***	0.29 (1.63–5.98) ***
Difficulty in Adapting to the Medical Team				
No	Ref.			Ref.
Yes	−0.09 (−2.59–0.51)			−0.07 (−2.50–0.75)
Difficulty in Performing Nursing Process				
No	Ref.			Ref.
Yes	−0.06 (−2.33–0.89)			−0.08 (−2.80–0.89)
Difficulty in Performing Nursing Skills				
No	Ref.			Ref.
Yes	0.00 (−1.58–1.59)			−0.07 (−2.65–0.96)

** *p* ≤ 0.01; *** *p* ≤ 0.001; △R^2^ = 0.033 in Model 1; △R^2^ = 0.093 in Model 2; △R^2^ = 0.009 in Model 3.

**Table 3 ijerph-19-08374-t003:** Comparison of the anxiety score in clinical practicum and the 5-year nursing professional employment of in-service participants in 2021 according to the emotional stability level (*n* = 202).

Emotional Stability (*n*)	Pre-Practicum BAI Score ≥ 8 (*n*) (%)	Post-Practicum BAI Score ≥ 8 (*n*) (%)	5-Year Nursing Professional Employment (*n*) (%)
Low (62)	Yes (25) (40.32%)	Yes (7) (11.29%)	Yes (3) (4.84%)
			No (4) (12.90%)
		No (18) (29.03%)	Yes (7) (11.29%)
			No (11) (14/62)
	No (37) (59.68%)	Yes (3) (4.84%)	Yes (0) (0)
			No (3) (4.84%)
		No (34) (54.84%)	Yes(13) (20.97%)
			No (21) (33.87%)
Medium (99)	Yes (13) (13.13%)	Yes (8) (8.08%)	Yes (4) (4.04%)
			No (4) (4.04%)
		No (5) (5.05%)	Yes (1) (1.01%)
			No (4) (4.04%)
	No (86) (86.16%)	Yes (11) (11.11%)	Yes (5) (5.05%)
			No (6) (6.06%)
		No (75) (75.76%)	Yes (32) (32.32%)
			No (43) (43.43%)
High (41)	Yes (3) (7.32%)	Yes (1) (2.43%)	Yes (1)(2.43%)
			No (0) (0)
		No (2) (4.88%)	Yes (0) (0)
			No (2) (4.88%)
	No (38) (92.68%)	Yes (2) (4.88%)	Yes (1) (2.43%)
			No (1) (2.43%)
		No (36) (87.80%)	Yes (13) (31.71%)
			No (23) (56.10%)

**Table 4 ijerph-19-08374-t004:** Summary of 5-year nursing professional employment of low emotional stability with the estimating variable (*n* = 62).

Estimating Variable		Estimate Condition on 5-Year Nursing Professional Employment	
Pre-Practicum BAI Score ≧ 8	Post-Practicum BAI Score ≧ 8	Failure or Not	Outcome and What the Participants Learned from Clinical Practicum
Yes	Yes	Failure; could not adapt anxiety after enrolling clinical setting.	There were 42.86% not failure.
	No	Not failure; could adapt anxiety after enrolling clinical setting.	There were 38.89% not failure.
No	Yes	Failure; could not adapt anxiety after enrolling clinical setting.	All were failure.
	No	Trace; could adapt anxiety after enrolling clinical setting.	There were 38.24% not failure.

## Data Availability

The data presented in this study are available upon request from the corresponding author. The data are not publicly available due to privacy.

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
