# Peer review of "The Influence of Reducing Clinical Practicum Anxiety on Nursing Professional Employment in Nursing Students with Low Emotional Stability"

_ijerph, 2022, doi:10.3390/ijerph19148374_

Round 1
Reviewer 1 Report
I would have liked to see more of the participants' comments and a fuller integration of the quantitative and qualitative elements of the study
Reviewer 2 Report
Thank you for the opportunity to review this study entitled “The Influence of Reducing Clinical Practicum Anxiety on Nursing Professional Employment in Nursing Students with Low Emotional Stability” (ijerph-1792038).
The study focused on the nursing students' experience of anxiety, exploring the factors influencing anxiety differences between pre-and post-practicum, idenfying their anxiety events in a clinical environment, and exploring the correlation between emotional stability and 5-year nursing professional employment. A sample of 237 nursing students was involved in the research.
In my opinion, the research topic is relevant, and the study is interesting. Parallelly, some issues need to be addressed before the paper will be suitable for publication.
1. Abstract: the information about the sample should be deepened (Mean age and SD? Percentage of men and women?) to provide a clear picture of what will be presented in the paper.
2. Introduction: “Based on a review of the literature, the anxiety events experienced by nursing students in clinical environments can be classified into three categories: difficulty in adapting to the medical team, difficulty in performing nursing processes, and difficulty in performing nursing skills.” Please provide the reference for the Review of literature on which this important sentence is based.
3. “Therefore, this study aimed to find if the emotional stability of nursing students can affect the clinical-practicum anxiety and the nursing employment after 5 years graduate?” Is a question? Do the authors ask the reader about the aims of their research?
4. “applied a secondary analysis method and quantitative approach” This sentence seems to be incomplete.
5. Complementary to the limitations, directions for future research should be indicated.
Reviewer 3 Report
It was with great pleasure that I did your review. Relevant topic for the present, taking into account what is recommended in the standards of learning and education of future nursesHere are some improvement suggestions,namely in the keywords that are not included in the descriptors used in the health area.
The implications of your study,for both students and training and how we can as teachers improve learning and the promotion of healthy schools fell a little short of expectations. Congratulations and best of luck for the revision of your manuscript
